

# A non-intrusive, multi-scale, and flexible coupling interface in WRF

Sébastien Masson[1], Swen Jullien[2], Eric Maisonnave[1], David Gill[3], Guillaume Samson[4], Mathieu Le Corre[5], Lionel Renault[6]

[1]LOCEAN-IPSL, Sorbonne Université (UPMC)/IRD/CNRS/MNHN, UMR7159, Paris, 75005, France
[2]Univ Brest, Ifremer, CNRS, IRD, LOPS, F-29280 Plouzané, France
[3]St Pius X (MS, Inc.), Aurora, 80011, USA
[4]Mercator Ocean International, Toulouse, 31400, France
[5]Service Hydrographie et Océanographie de la Marine, Brest, France
[6]LEGOS, Université de Toulouse, CNES-CNRS-IRD-UPS, 31400, France

*Correspondence to*: Sébastien Masson (sebastien.masson@locean.ipsl.fr)

**Abstract.** The Weather Research and Forecasting (WRF) model has been widely used for various applications, especially for solving mesoscale atmospheric dynamics. Its high-order numerical schemes and nesting capability enable high spatial resolution. However, a growing number of applications are demanding more realistic simulations through the incorporation

of coupling with new model compartments and an increase in the complexity of the processes considered in the model. (e.g., ocean, surface gravity wave, land-surface, chemistry...). The present paper details the development and the functionalities of the coupling interface we implemented in WRF. It uses the Ocean-Atmosphere-Sea-Ice-Soil - Model Coupling Toolkit (OASIS3-MCT) coupler, which has the advantage of being non-intrusive, efficient, and very flexible to use. OASIS3-MCT has already been implemented in many climate and regional models. This coupling interface is designed with the following

baselines: (1) it is structured with a 2-level design through 2 modules: a general coupling module, and a coupler-specific module, allowing to easily add other couplers if required, (2) variables exchange, coupling frequency, and any potential time and grid transformations are controlled through an external text file, offering great flexibility, (3) the concepts of "external domains" and "coupling mask" are introduced to facilitate the exchange of fields to/from multiple sources (different models, fields from different models/grids/zooms...). Finally, two examples of applications of ocean-atmosphere coupling are

proposed. The first is related to the impact of ocean surface current feedback to the atmospheric boundary layer, and the second concerns the coupling of surface gravity waves with the atmospheric surface layer.

**Short Summary.** This article details a new feature we implemented in the most popular regional atmospheric model (WRF). This feature allows data to be exchanged between WRF and any other model (e.g. an ocean model) using the coupling

library Ocean-Atmosphere-Sea-Ice-Soil - Model Coupling Toolkit (OASIS3-MCT). This coupling interface is designed to be non-intrusive, flexible and modular. It also offers the possibility of taking into account the nested zooms used in WRF or in the models with which it is coupled.



# 1 Introduction

The Weather Research and Forecasting (WRF, Skamarock 2004) model is probably the most popular atmospheric regional
model with more than 50,000 users in 160 countries. This state-of-the-art non-hydrostatic atmospheric model is used in a
wide range of atmospheric research and operational forecasting applications at scales ranging from thousands of kilometers
to tens of meters. WRF's success can be explained in many ways (easy configuration setup, a large and active community,
numerical performances, ...), but the key feature of this model is definitely the quality of its results thanks to the extended
choice of available physical parameterizations, and to its dynamic solver (ARW, Skamarock and Klemp 2008) that was
especially designed with high-order numerical schemes to enhance the model's effective resolution of mesoscale dynamics.
Another popular feature of WRF is its capability for nesting, which allows running a part of the model domain at higher
spatial resolution. These nests can be defined either at the same level (sibling nests) or nested within each other to any depth
(parent-children nests). The position of the nest within the parent grid can be fixed in time or can move, either along a
specified trajectory or following a predefined criterion (e.g., low in the 500 mb height).


The WRF community continually proposes new contributions to improve and/or add features to the model. One way of
improving the quality and realism of numerical simulations is to refine the representation of physical processes in the model.
This can be achieved by increasing the level of details and complexity of modelled processes or incorporating new processes
or even new model compartments in the system. Following the example of the climate modelling community, which started
to use coupled models more than 50 years ago (Manabe and Bryan, 1969), there is a growing number of applications using
regional atmospheric models such as WRF coupled to another model such as ocean, surface gravity wave, land-surface or
chemistry. WRF has been coupled to numerous ocean models, notably the Coastal and Regional Ocean Community Model
(CROCO, Renault et al. 2019a), the MIT General Circulation Model (MITgcm, Sun et al. 2019), the Nucleus for European
Modelling of the Ocean (NEMO, Samson et al. 2014), the Princeton Ocean Model (POM, Liu and al 2011), the Hybrid
Coordinate Ocean Model (HYCOM, Chen et al. 2013), the Parallel Ocean Program (POP, Cassano et al. 2017), or the
Regional Ocean Modeling System (ROMS, Warner et al. 2010).

Coupling WRF to another model can simply be achieved by exchanging data through files (e.g., Jullien et al. 2014), however
most coupled models nowadays use a coupler, which allows direct data exchange, offering better performances and more
flexibility, particularly with regards to grid interpolation. Today, WRF is therefore coupled using the most common couplers
as for example: the Earth System Modeling Framework (ESMF, Hill et al. 2004), the Community Coupler (Liu et al., 2014),
the Model Coupling Toolkit (MCT, Larson et al. 2005), either directly or through CPL7 (Craig et al. 2012), or through the
Ocean-Atmosphere-Sea-Ice-Soil (OASIS3-MCT version 5, Craig et al. 2017) as detailed in the present paper. Each coupler
has its own benefits and drawbacks, with none being universally suitable for all constraints, requirements, and practices of
the various groups that employ them. Valcke (2022) classifies them into two main categories: the "external coupler or
coupling library" (typically C-Coupler and OASIS3-MCT) and the "integrated coupling framework" (typically ESMF and
CPL7). WRF incorporates a coupling interface with one representative from each of these two categories: ESMF (already in





WRF version 3) and the OASIS3-MCT (since version 3.6 in 2014). The main objective of this publication is to provide an extensive description and user guide of this OASIS-MCT coupling interface. The update of this interface, phased with WRF
4.6.0, has motivated the writing of this paper, which fills the gap in the documentation of this work we initiated a decade ago.

The OASIS3-MCT coupler was designed to easily couple various models with minimal changes required to the models being coupled. Often described as the "Swiss Army Knife" coupler, OASIS3-MCT is a set of libraries, which allows to
exchange variables between different models and perform grid interpolations and time transformations if requested by the user (see OASIS3-MCT user guide for all details, Valcke et al. 2021). OASIS3-MCT is fully parallelized (thanks to the MCT engine), ensuring good computational performance. It has the advantage of being non-intrusive (only a few calls in the model time stepping, and a few additional calls for communicators, grids and sub-domains' definition) and very flexible to use. Once the coupling interface has been implemented into the code, the users can define their coupling strategy (which
variables are exchanged with what spatial and temporal treatment) directly through an external text input file, allowing for flexibility without requiring any additional adjustments to the source code. The qualities of OASIS3-MCT explain its high popularity and its use in 7 of CMIP6 global climate models as well as in various components of regional models (see examples at https://oasis.cerfacs.fr/en/results-of-the-survey-2019-on-oasis3-mct-coupled-models/). The de facto format that OASIS provides facilitates the coupling of any model component that already integrates an OASIS based interface. WRF has
thus been coupled through this interface with various models, including the chemistry-transport model CHIMERE (Briant et al. 2017), the land surface model ORCHIDEE (Guion et al. 2022), the surface gravity wave model WAVEWATCH III (Tolman, 2009), or the previously mentioned ocean models CROCO and NEMO.

The ocean-atmosphere coupling is by far the most popular application of this work, which has already been used in
numerous studies. Many of them showed the importance of the air-sea coupling at oceanic (sub)mesoscale in various regions: the Agulhas current (Renault and al. 2017), the Bay of Bengal (Krishnamohan et al. 2019), the California (Renault and al. 2016a, 2018), the English Channel (Renault and Marchesiello 2022), the Gulf of Mexico (Larrañaga et al. 2022), the Gulf Stream (Renault and al. 2016b, 2019), the Mediterranean Sea (Renault et al. 2021), the south-eastern Pacific (Oerder et al. 2016, 2018), the Tropical Atlantic (Gévaudan et al. 2021), or even the entire tropical channel (Jullien et al. 2020, Renault
et al. 2019c, 2020, 2023). Other studies used this coupling interface to focus on tropical cyclones (e.g., Samson et al. 2014, Lengaigne et al. 2019, Neetu et al. 2019) or the Indian Monsoon (Samson et al. 2017, Terray et al. 2018). The Model of the Regional Coupled Earth System (MORCE, Drobinski et al., 2012) platform is also benefiting from this coupling interface. MORCE was used in diverse projects such as Med-CORDEX (Ruti et al., 2016) with application on the local atmospheric dynamic (Drobinski et al., 2018) or extreme meteorological events (Lebeaupin Brossier et al. 2015, Berthou et al. 2016,
Panthou et al. 2018).





The present paper describes in detail the implementation and the usage of the OASIS3-MCT coupling interface into WRF. In section 2, the general philosophy of the non-intrusive and flexible interface is given. Its detailed implementation in the code, the changes to the original code to add the coupling interface, as well as the few modifications needed to activate the coupling interface at the compilation stage are detailed. In section 3, different applications of the interface are illustrated, along with the few additional changes to the WRF original code. Finally in section 4, the ability of this tool to go towards multi-scale applications is exposed, with the implemented concept of "coupling mask" allowing to couple various domains including or not embedded zooms.

## 2 A non-intrusive and flexible coupling interface

The design of this coupling interface was motivated by the idea of limiting modifications to the original WRF code as much as possible, in order to facilitate its maintainability. To do so we used the OASIS3-MCT coupler (Craig et al. 2017), which requires very few intrusions into the code, and we adopted a few coding rules:

- Isolate the interface itself in dedicated new modules: "module_cpl.F" and "module_cpl_oasis3.F" (which are detailed in section 2.1)
- Limit the modifications to the original code by only adding calls to coupling subroutines
- Distinguish these coupling subroutines with a name starting with "cpl_"
- Mark off and control the calls to these subroutines with a test on a logical, named "coupler_on"

The implementation will be fully detailed in section 2.4, but let's first introduce the overall philosophy of our coupling strategy.

## 2.1 A two-level coupling interface

The coupling interface was written to be used with OASIS3-MCT. However, one could argue that the main steps of the coupling are quite generic and applicable to most couplers. We therefore structured our coupling interface with a 2-level design through 2 modules: a generic coupling module, "frame/module_cpl.F", and a dependency module for the chosen coupler, "frame/module_cpl_oasis3.F".

The first module "frame/module_cpl.F" gathers all the subroutines called in the other parts of the code. This module is thus the only coupling module "used" in the other original WRF routines (i.e. with the Fortran instruction "USE module_cpl"). Its public subroutines constitute the set of generic actions that should be required by the coupler. This module is always compiled, even if the user is not doing any coupling, along with other WRF modules. We tried to limit the use of C preprocessor keys, which tend to create "dead code" over time. This also makes the code easier to read by limiting the





preprocessing command lines and ensures that the coupling interface is compiled when new modifications are made to the
code (and therefore checked obvious bugs).

In this module, we define the logical "coupler_on" and the string of characters "coupler_name" which defines the coupler we
are using, as shown in Figure 1.

```
#ifdef key_cpp_oasis3
    LOGICAL      , PARAMETER, PUBLIC :: coupler_on   = .TRUE.
    CHARACTER(5), PARAMETER          :: coupler_name = 'oasis'
#else
    LOGICAL      , PARAMETER, PUBLIC :: coupler_on   = .FALSE.
    CHARACTER(4), PARAMETER          :: coupler_name = 'none'
#endif
```

**Figure 1: Declaration of "coupler_on" and "coupler_name" in frame/module_cpl.F.**

As of today, the only implemented coupler using this interface is OASIS3-MCT. The choice in the definition of
coupler_name is thus limited to 2 cases: 'none' or 'oasis', but the structure of "frame/module_cpl.F" is designed to add more
choices.

OASIS3-MCT specific interface is isolated and defined in a separate new module: "frame/module_cpl_oasis3.F". This
second module is the only location where we use OASIS3-MCT routines. It is thus the only file containing the following call
to OASIS module:

USE mod_oasis   ! OASIS3-MCT module

The public routines of frame/module_cpl_oasis3.F are only used in "frame/module_cpl.F", which minimizes the intrusion of
OASIS3-MCT into WRF code. The use of the C preprocessor key "key_cpp_oasis3" ensures that this routine can be
compiled without "key_cpp_oasis3" and generates, in this case, dummy routines allowing the compilation of
"frame/module_cpl.F".

**2.2 The coupling sequence**

The coupling sequence (Fig. 2) is structured into 3 steps with functionalities corresponding to specific Fortran subroutines
callable from the original code:

- Initialization and definition phase: take care of MPI communicators and MPI sub-domains definition
- Temporal loop and exchange phase: potentially receive/send data from/to the coupler at the beginning/end of each
time step
- End of simulation: finalize or abort coupling

The Initialization, definition, and finalization steps are done once in each model outside of the temporal loop, while the
send/receive interfaces are called at every time step (Fig. 2). Nevertheless, the effective exchanges of data between two
models are done at the coupling time step that is usually larger than the models time step (e.g 1 hour). This coupling time
step, which must be a common multiple of each model time step, is defined by the user in an external text file read by the





coupler (see next sub-section). An example of coupling sequence is illustrated in Figure 3 featuring two models with distinct time steps, dt$_1$ and dt$_2$. In this example, the coupling time step is 2 times dt$_1$ and 4 times dt$_2$.




**Figure 2: Schematic view of the coupling steps implemented in WRF coupling interface: initialization, definition, exchanges, finalization. Here is an example of 2 models coupled through OASIS3-MCT is illustrated.**



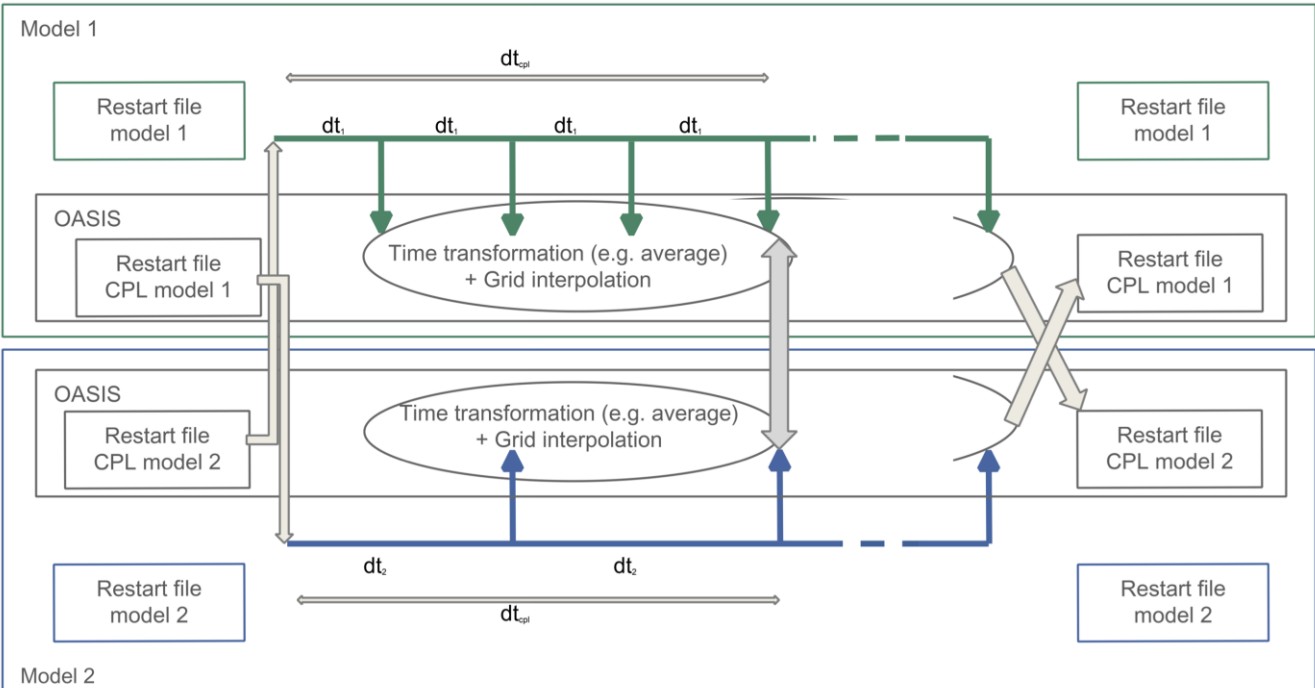

**Figure 3: Schematic view of the coupling sequence used in a coupled simulation between 2 models interfaced using the OASIS3-MCT library.**

## 2.3 A coupling strategy controlled by an external file

The coupling interface is further configured through a simple external text file which allows to set up the coupled simulation without modifying or recompiling the code. With OASIS3-MCT coupler, this file is called the "namcouple". It allows to completely configure the coupled simulation by specifying:

- Which variable will be exchanged,
- To/from which domain,
- At which frequency,
- With which temporal treatment and spatial interpolation.

The list of variables potentially sent or received by WRF is hard coded in the subroutine "cpl_init" of "frame/module_cpl.F". However, this does not necessarily mean that they will be used or coupled. The variables to be coupled are selected in the namcouple file and, in WRF, for each potential coupling variable, we check whether it is actually required in the user-defined namcouple. Each coupling variable is identified through a name, which is hardcoded and stored in a character array: named either "rcvname" (received) or "sndname" (sent) and defined as private variables of the module "frame/module_cpl.F". The maximum length of rcvname or sndname names is arbitrary defined to 64 characters. The current list of these potentially exchanged variables will be detailed in section 3.3.





The coupling subroutines designed for the exchanges ("cpl_tosend" and "cpl_toreceive" in "frame/module_cpl.F") are called at every time-step which is, for example, needed to compute time average or check if it is time to receive some data. Note that, when sending data, all time transformations are performed locally by the model sending data without requiring any MPI communication. The effective exchange of data between models, involving MPI communications, is performed only when the domain integration reaches a coupling time-step defined independently for each exchanged variable in the namcouple (Fig 3).

The time transformations and grid interpolation methods available in OASIS3-MCT are summarized in Tables 1 and 2, see also OASIS3-MCT user guide for further details (Valcke et al. 2021). All intermediate arrays needed for some of the time transformation (e.g.. average) are managed internally and automatically by OASIS3-MCT without any additional code lines in WRF. In the OASIS3-MCT namcouple, users have the option to determine whether spatial interpolations should be applied to time-transformed data by the sending or the receiving model. This decision provides greater flexibility in optimizing the load balance of the models. OASIS3-MCT can automatically compute interpolation weights for certain spatial interpolations based on input files specifying the grid characteristics (see Valcke et al. 2021 for all details). These input files, called "grids.nc", "masks.nc" and "areas.nc", can be automatically built from the WRF "geogrid" files (i.e "geo_em.dxx.nc", where xx is WRF domain number) using the shell script in Appendix 1. Finally, OASIS3-MCT uses a dedicated restart file for each model. Since our model uses quantities accumulated during the previous coupling step, the OASIS restart file contains the initial or restart fields that will be used to initiate or restart a simulation. At the end of the run, OASIS3-MCT automatically writes the new restart files to be used at the start of the next chunk of the simulation.

| INSTANT | no time transformation, the instantaneous field is transferred |
|---------|---------------------------------------------------------------|
| ACCUMUL | the field accumulated over the previous coupling period is exchanged |
| AVERAGE | the field averaged over the previous coupling period is transferred |
| T_MIN | the minimum value of the field for each source grid point over the previous coupling period is transferred |
| T_MAX | the maximum value of the field for each source grid point over the previous coupling period is transferred |

**Table 1: Time transformations available in OASIS3-MCT using the LOCTRANS keywork in the namcouple. See OASIS3-MCT user guide for more detailed information (Valcke et al. 2021).**

| BILINEAR, BILINEARNF | interpolation based on a local bilinear approximation with/without a nearest neighbor fill for non-masked target points that do not receive a value because all the 4 source grid points are masked. |
|---------------------|-------------------------------------------------------------------------------------------|
| BICUBIC, BICUBICNF | interpolation based on a local bicubic approximation with/without a nearest neighbor fill for non-masked target points that do not receive a value. |



| CONSERV | 1st or 2nd order conservative remapping |
|---|---|
| LOCCUNIF, LOCCDIST and LOCCGAUS | Locally conservative interpolation by associating N target nearest neighbors to every SOURCE grid point and applying a weight normalization considering the source/target mesh area ratio. |
| DISTWGT, DISTWGTNF | distance weighted nearest-neighbor interpolation (N neighbors) with/without a nearest neighbor fill for non-masked target points that do not receive a value. |
| GAUSWGT, GAUSWGTNF | N nearest-neighbor interpolation weighted by their distance and a gaussian function with/without a nearest neighbor fill for non-masked target points that do not receive a value. |
| MAPPING | Any user defined interpolation file following the SCRIPR format (https://github.com/SCRIP-Project/SCRIP) |

**Table 2: Spatial interpolation available in OASIS3-MCT. See OASIS3-MCT user guide for more detailed information (Valcke et al. 2021).**

## 2.4 Detailed implementation in WRF

In the following, we detail the implementation of the coupling interface in WRF. A schematic representation is also depicted in Figure 4.

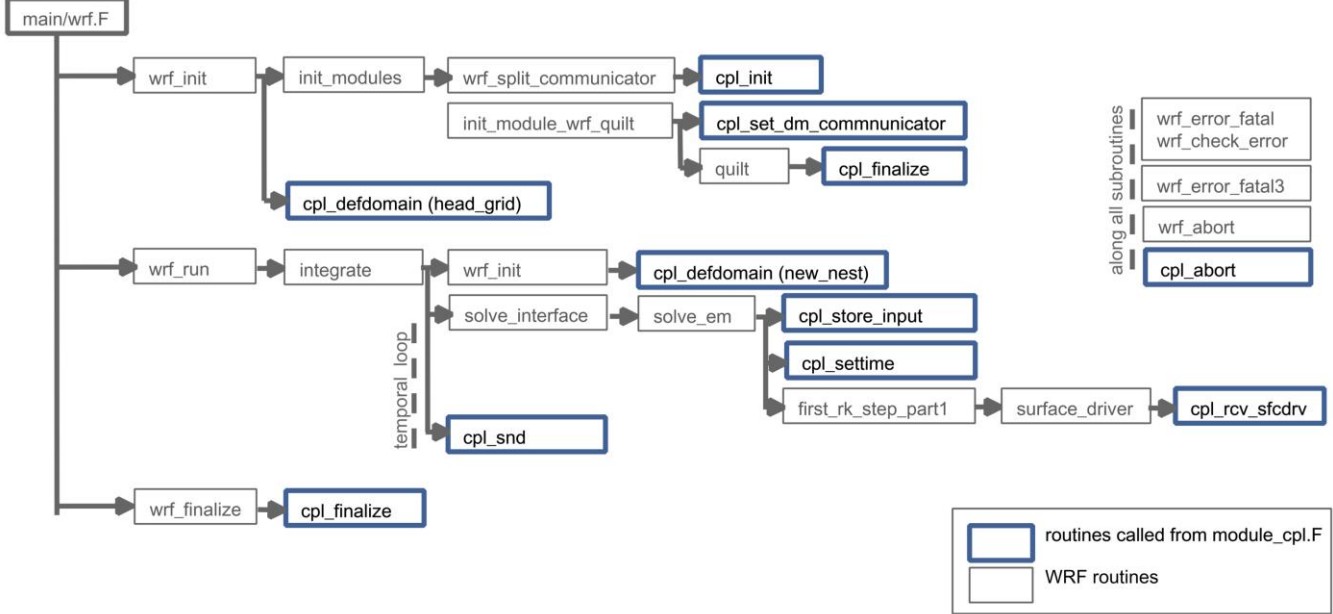

**Figure 4: Schematic view of the coupling interface implementation in WRF. WRF original routines are in grey. All new routines**
**(in blue) are gathered in "frame/module_cpl.F".**



### 2.4.1 Initialization phase

**Defining the MPI communicator.**

The first task of the coupler is to handle the MPI communicator. This is done in the WRF split_communicator subroutine of "external/RSL_LITE/module_dm.F". WRF is using the variable "mpi_comm_here" as a global communicator, which is
defined to MPI_COMM_WORLD by default. When coupling, this MPI communicator is defined by calling the "cpl_init" subroutine, as shown in Figure 5:

```
IF ( coupler_on ) THEN
    CALL cpl_init( mpi_comm_here )
ELSE
    CALL mpi_init ( ierr )
    mpi_comm_here = MPI_COMM_WORLD
END IF
```

**Figure 5: Modification of "external/RSL_LITE/module_dm.F" to add the call to "cpl_init". Lines added for the coupling interface are in blue. Original code lines are in black.**

When using OASIS3-MCT as the coupling library, mpi_comm_here is given by OASIS3-MCT, which defines a local communicator for each model, including all the model processes involved in the coupling. The MPI_COMM_WORLD communicator is reserved by OASIS to properly close the simulation. It is also possible to couple through OASIS3-MCT with all components gathered in a single executable. In this case, OASIS3-MCT split the single executable communicator into several communicators, each of them addressing one component. The current implementation of the coupling interface
is following the first strategy which corresponds to the usual usage of OASIS3-MCT which is less intrusive.

An additional step is required when WRF is using its IO quilting. We must indeed specify to the coupler which MPI tasks are dedicated to the model integration (the "compute nodes"), and which are dedicated to the IO quilting (the "server nodes"). This code modification is done in WRF "init_module_wrf_quilt" subroutine of "frame/module_io_quilt_old.F" or
"frame/module_io_quilt_new.F". Compute nodes send to the coupler their local communicator, mpi_comm_local (defined by "wrf_set_dm_communicator"), by calling the "cpl_set_dm_communicator" subroutine, whereas server nodes send the MPI_COMM_NULL communicator to specify to the coupler that they are not included in the coupling, as shown in Figure 6.

```
IF ( compute_node ) THEN
    IF (coupler_on) CALL cpl_set_dm_communicator( mpi_comm_local )
ELSE
    IF (coupler_on) CALL cpl_set_dm_communicator( MPI_COMM_NULL )
    mpi_comm_local = mpi_comm_local_io_server_tmp
    CALL quilt    ! will not return on io server tasks
ENDIF
```

**Figure 6: Modification of "frame/module_io_quilt_old.F" to add the call to "cpl_set_dm_communicator". Lines added for the coupling interface are in blue. Original code lines are in black.**





**Mapping the MPI subdomains**

Once the proper MPI communicators are defined, the next step is to provide to the coupler the mapping of the MPI subdomains, that is identifying which compute node is dedicated to which part of the model grid integration. In WRF, the
domain decomposition is defined in "alloc_and_configure_domain", which is called at 2 different places in the code: "wrf_init" subroutine in "main/module_wrf_top.F" for the head grid and integrate subroutine in "frame/module_integrate.F" for the nested child grids. In both cases, the "cpl_defdomain" subroutine is used to provide the grid definition to the coupler. For the parent grid, the call to "cpl_defdomain" is done at the end of the "wrf_init" subroutine, as shown in Figure 7.

```
      IF (coupler_on) CALL cpl_defdomain( head_grid )
    ENDIF  ! domain_active_this_task
END SUBROUTINE wrf_init
```

**Figure 7: Modification of "main/module_wrf_top.F" to add the call to "cpl_defdomain". Lines added for the coupling interface are in blue. Original code lines are in black.**

For the child grids, the call to cpl_defdomain is done at the end of the while loop on the number of children at the beginning of integrate, as shown in Figure 8.

```
DO WHILE ( nests_to_open( grid , nestid , kid ) )
...
    CALL alloc_and_configure_domain ( domain_id  = nestid ,   &
                                      grid       = new_nest , &
...
      IF (coupler_on) CALL cpl_defdomain( new_nest )
    ENDIF ! active_this_task
END DO
```

**Figure 8: Modification of "frame/module_integrate.F" to add the call to "cpl_defdomain". Lines added for the coupling interface are in blue. Original code lines are in black.**

In addition to the grid partitioning, the "cpl_defdomain" subroutine determines the variable exchange between parent or child domains. This selection of the coupled variables is further detailed in section 3.3.

Note that OASIS3-MCT imposes that all grid and MPI partitioning definitions are done before starting exchanges from/to a given model. This constrain has three consequences regarding the use of nested grids in coupled mode:

- First, in coupled mode with OASIS3-MCT, all the nested grids must be instantiated and ceased at the same time of the parent grid. This is not required in WRF stand-alone mode.

- Second, as the grid definition must be done only once and before any variable exchange, moving nests can be used
but cannot be directly coupled. As detailed in the discussion, they can however be coupled indirectly by coupling only the static parent domain on which the moving nests feedback.

- Third, as presently implemented in WRF, the coupling interface works with a maximum of 1 level of nested grids. Indeed, in WRF, the definition of the child domains is done at the beginning of the first time step of the parent grid (notably to allow different start/stop dates for nested grid, as mentioned in the first point). Thus, the first level of





child grid (whose parent is head_grid) is defined before any exchange, which is ok for OASIS3-MCT. However, as
the integrate subroutine is recursive, the second level of child grid is defined after the first step of the grand-parent
grid, which is not allowed by OASIS3-MCT. A solution to this limitation is proposed in the discussion.

### 2.4.2 Temporal loop

The temporal loop is performed within the recursive integrate subroutine in "frame/module_integrate.F" with a while loop.
The time integration of a single time step is ensured by a call to the solver ("dyn_em/solve_em.F" in our case), which is
applied in a do loop on each sibling domain (i.e., domains at the same level in the zooms hierarchy), as shown inf Figure 9.

```
grid_ptr => grid
DO WHILE ( ASSOCIATED( grid_ptr ) )
   ...
   CALL solve_interface ( grid_ptr )
   ...
   grid_ptr => grid_ptr%sibling
END DO
```

**Figure 9: recursive integration of the different nest in "frame/module_integrate.F".**

**Initialization part of the solver**

Two calls to the coupling routines are performed in the initialization part of the solver before the call to the first part of the
Runge-Kuta scheme. We first call the "cpl_store_input" subroutine, which eventually copy data that has been read in
AUXINPUT4 input file ("wrflowinp_dxx" file with xx the domain number) in order to keep a copy of the wrflowinp values
before the reception of the corresponding data from the coupler (see section 3.1.2 for additional information):

290        IF (coupler_on) CALL cpl_store_input( grid, config_flags )

The "cpl_store_input" subroutine requires an update of the variable "just_read_auxinput4" at the end of the "solve_em"
subroutine to specify if new data has just been read in the AUXINPUT4 input file.

        IF (coupler_on) grid%just_read_auxinput4 = Is_alarm_tstep(grid%domain_clock,
        grid%alarms(AUXINPUT4_ALARM))

The second call to coupling routines before the actual integration is the call to "cpl_settime", which provides to the coupler
the time (in seconds) since the beginning of the simulation (from cold or hot restart):

        IF (coupler_on) CALL cpl_settime( curr_secs2 )

**Exchanging variables**

Up to now, all coupling fields defined in the coupling interface are "surface" data (2D-arrays) sent by an "external domain"
(i.e. another domain grid than WRF d01, d02...) belonging for example to an ocean, a wave or a land model. These "surface"
data must be received before the calls to the surface parameterizations. The reception of the data sent by the coupler must
thus be done at the beginning of the "surface_driver" subroutine of "phys/module_surface_driver.F". This is done with a
simple call to the coupling subroutine named "cpl_rcv_sfcdrv" (for surface driver), as shown in Figure 10.





```
IF ( coupler_on ) THEN
   CALL cpl_rcv_sfcdrv( id, max_edom, cplmask, cosa, sina,    &
       &                SST_INPUT, SST, UOCE, VOCE,   &
       &                CHA_COEF,                      &
       &                ids, ide, jds, jde, kds, kde, &
       &                ims, ime, jms, jme, kms, kme, &
       &                ips, ipe, jps, jpe, kps, kpe  )
END IF
```


**Figure 10: Call to "cpl_rcv_sfcdrv" that was added in"phys/module_surface_driver.F". Lines added for the coupling interface are in blue.**

The surface data SST, UOCE, VOCE, and CHA_COEF are the outputs parameters (see section 3.3 for details) of the "cpl_rcv_sfcdrv" subroutine, whereas all other parameters are "intent(in)".

For each sibling, a recursive call to the integrate subroutine is ensuring that all childrens are also proceeding their temporal integration by calling "solve_em".

Coupling fields are then sent to the coupler at the end of the integrate subroutine, once the parent grid time step and the child grids sub-time steps integrations have been performed, and once all child domains have feedback to their parent. The "cpl_snd" subroutine is called to achieve this task, as shown in Figure 11.

```
grid_ptr => grid
DO WHILE ( ASSOCIATED( grid_ptr ) )
   ...
   DO kid = 1, max_nests
      ...
      CALL integrate ( grid_ptr%nests(kid)%ptr )
      ...
   ENDDO
   ...
   DO kid = 1, max_nests
      ...
      CALL med_nest_feedback ( grid_ptr, grid_ptr%nests(kid)%ptr, config_flags )
      ...
   END DO
   ...
   IF (coupler_on) CALL cpl_snd( grid_ptr )
   grid_ptr => grid_ptr%sibling
END DO
```


**Figure 11: Schematic representation of the while loop on the different nests in "frame/module_integrate.F" with the added call to "cpl_snd". Lines added for the coupling interface are in blue. Original code lines are in black.**





### 2.4.3 End of the simulation

In coupled mode, at the end of the simulation, when reaching the last lines of the "wrf_finalize" subroutine in "main/module_wrf_top.F", the "cpl_finalize" subroutine is called instead of "WRFU_Finalize" and "wrf_shutdown" in stand-alone mode, as shown in Figure 12.

```
IF (coupler_on) THEN
    CALL cpl_finalize()
ELSE
    CALL WRFU_Finalize
    CALL wrf_shutdown
ENDIF
```

**Figure 12: Added call to "cpl_finalize" in "main/module_wrf_top.F". Lines added for the coupling interface are in blue. Original code lines are in black.**

The "wrf_abort" subroutine in "external/RSL_LITE/module_dm.F" has also been modified to allow in case of error, a clean abort of WRF in the coupling interface, and a clear associated error message. This prevents for any deadlock in the coupling interface. The dedicated "cpl_abort" subroutine is thus called in coupled mode instead of the usual "mpi_abort", see Figure 13.

```
IF ( coupler_on ) THEN
    CALL cpl_abort( 'wrf_abort', 'look for abort message in rsl* files' )
ELSE
    CALL mpi_abort(MPI_COMM_WORLD,1,ierr)
END IF
```

**Figure 13: Added call to "cpl_abort" in "external/RSL_LITE/module_dm.F". Lines added for the coupling interface are in blue. Original code lines are in black.**

### 2.5 Compilation

As detailed previously, our coupling interface is structured with a 2-level design separating generic coupling routines, and coupler-specific routines. This allows coupling WRF with different couplers. At the current stage, we have only interfaced the "OASIS3-MCT" coupler, but the procedure should be similar with couplers which requires the same kind of input than OASIS, such as YAC or C-Coupler.

OASIS3-MCT is a set of libraries that has to be downloaded and compiled before compiling WRF (with the same compiler
and preferably with the same compiler version, see https://portal.enes.org/oasis for details), and then linked at the end of WRF compilation. Activating the OASIS3-MCT coupling interface in WRF thus requires a dedicated WRF compilation with a few changes in the configure.wrf file:

- First, the additional C preprocessor key named "key_cpp_oasis3" must be added to the list of keys already defined in the ARCH_LOCAL variable:

345             ARCH_LOCAL      = -Dkey_cpp_oasis3 –D...





- Second, the paths to OASIS3-MCT include and library directories must be added, so that the compiler knows where to find them. The simplest way to proceed is to follow what is done, for example, for the treatment of NetCDF include and library paths in configure.wrf. First, we define an additional variable OA3MCT_ROOT_DIR, which defines the root directory for OASIS3-MCT:

350                          OA3MCT_ROOT_DIR = /…/oasis3-mct/BLD

Then, we add OASIS3-MCT paths to the list of include modules, as shown in Figure 14.

For OASIS3-MCT version 5:

```
INCLUDE_MODULES = $(MODULE_SRCH_FLAG) \
                  ...
                  -I$(WRF_SRC_ROOT_DIR)/inc \
                  -I$(OA3MCT_ROOT_DIR)/include
```

For OASIS3-MCT older versions:

```
INCLUDE_MODULES = $(MODULE_SRCH_FLAG) \
                  ...
                  -I$(WRF_SRC_ROOT_DIR)/inc \
                  -I$(OA3MCT_ROOT_DIR)/build/lib/mct \
                  -I$(OA3MCT_ROOT_DIR)/build/lib/psmile.MPI1
```

**Figure 14: Added path to OASIS3-MCT includes in configure.wrf. Added lines are in blue. Original lines are in black.**

Finally, we complete the list of library paths, keeping in mind that it is safer to respect the following rule in the order of the
libraries: "if library A depends on library B, library A must be listed before library B". As OASIS3-MCT depends on NetCDF, the OASIS3-MCT libraries must be listed before NetCDF libraries, as shown in Figure 15.

```
LIB_EXTERNAL    = \
                  -L$(WRF_SRC_ROOT_DIR)/external/io_netcdf -lwrfio_nf \
                  -L$(OA3MCT_ROOT_DIR)/lib -lpsmile.MPI1 -lmct -lmpeu -lscrip \
                  -lnetcdff -lnetcdf -lhdf5_hl -lhdf5 -lz -lcurl
```

**Figure 15: Added path to OASIS3-MCT libraries in configure.wrf. Added lines are in blue. Original lines are in black.**

Once these modifications of configure.wrf have been done, WRF can be compiled as usual.

**3 Ocean-Wave-Atmosphere coupling and exchanged variables**

This section details some of the coupling applications that could be done with the current coupling interface.

**3.1 Ocean-Atmosphere coupling and in particular ocean current coupling with modifications to the PBL schemes**

Since a decade or so, ocean-atmosphere interactions have been shown to have a large influence not only for the climate but also at smaller scales, such as oceanic mesoscale and submesoscale, with a rectifying effect at larger scale (Seo et al. 2023).





These interactions are mainly driven by two feedback mechanisms from the ocean to the atmosphere: the Thermal FeedBack
       (TFB), which is the influence of Sea Surface Temperature (SST) gradients and anomalies on the atmosphere; and the Current
       FeedBack (CFB), which is the influence of sea surface current on the atmosphere. Coupling the atmosphere to the ocean
       therefore involves the exchange of several fields. From the atmosphere to the ocean model, the surface heat, water, and
       momentum fluxes are sent. These only require some fields' transformation in the "cpl_snd" subroutine of "module_cpl.F"
(e.g., computing wind stress components, net heat fluxes), and does not imply any modifications to the original WRF
       routines. From the ocean to the atmosphere model, the SST, and the ocean currents components (UOCE, VOCE) are sent to
       the atmosphere. The SST coupling does not imply any changes to the WRF original routines, while the current feedback to
       the atmosphere requires a few modifications both in the surface driver and in the PBL parameterization routines (Renault et
       al. 2019b). Indeed, because of the implicit treatment of the bottom boundary condition, accounting for the relative motion of
the atmosphere and the ocean involves a modification of both the surface layer parameterization, and the tridiagonal matrix
       for vertical turbulent diffusion (Lemarié, 2015). In WRF, as the building of the tridiagonal system is done locally in each
       PBL parameterization, accounting for current feedback thus has to be done for each PBL parameterization. We have, for
       now, implemented the requested modifications in 2 PBL schemes: the Yonsei University (YSU, Hong et al. 2006) and the
       Mellor–Yamada Nakanishi Niino (MYNN, Nakanishi, M., and H. Niino, 2009) schemes. The required modifications are
summarized in Figure 16. These modifications have no implications in case of WRF stand-alone run, as the ocean current
       velocities (i.e., UOCE and VOCE variables) are set to 0 by default.





The changes in "module_surface_driver.F" are common to all schemes and read:

```
! remove surface currents for atmospheric low-level winds
        u_phytmp(i,kts,j)=u_phytmp(i,kts,j)-uoce(i,j)
        v_phytmp(i,kts,j)=v_phytmp(i,kts,j)-voce(i,j)
```

The changes in module_bl_ysu.F read:

```
    do i = its,ite
      wspd1(i) = sqrt( (ux(i,1)-uox(i))*(ux(i,1)-uox(i)) &
                     + (vx(i,1)-vox(i))*(vx(i,1)-vox(i)) )+1.e-9
    enddo
...
      ztmp = ust(i)**2*rhox(i)*g/del(i,1)*dt2/wspd1(i)*(wspd1(i)/wspd(i))**2
      f1(i,1) = ux(i,1)+uox(i)*ztmp
      f2(i,1) = vx(i,1)+vox(i)*ztmp
```

The changes to module_bl_mynn.F read:

```
      d(1)=u(k) + dtz(k)*uoce*ust**2/wspd - dtz(k)*s_awu(k+1)*onoff + &
          sub_u(k)*delt + det_u(k)*delt
```

**Figure 16: Modifications of "phys/module_surface_driver.F", "phys/module_bl_ysu.F" and "phys/module_bl_mynn.F" to include ocean currents coupling.**

### 3.2 Atmosphere-Wave coupling and modifications to the surface schemes

Atmosphere and surface gravity wave coupling may appear obvious while observing the ocean surface under various wind conditions. In atmospheric models, the wave interface and energy transfers at the air-sea interface are parameterized through a bulk formulation of air-sea fluxes (e.g., Charnock, 1955). However, the underlying mechanisms responsible for atmosphere-wave coupling continue to be a topic of ongoing discussion (e.g., Soloviev and Kudryavtsev 2010; Hristov 2018; Ayet et al. 2020). The effect of waves in bulk formulations is also considered as an average effect varying only with wind speed, while observations show that the sea state depends on numerous other factors (e.g., wave age, crossed seas, wave-currents interactions). One way to account for the variability related to sea-state is to incorporate a Charnock coefficient that is dependent on the sea-state in the bulk formulation. This can be achieved by either calculating it from a modelled wave spectrum (Janssen et al., 2001) or by using it as a function of key wave parameters, such as the wave age or steepness (e.g., Moon et al., 2004; Drennan et al., 2005) or wave height and mean wavelength (e.g. Warner et al. 2010).

Following this approach, we use the Charnock coefficient (CHA_COEF) computed in the wave model to compute roughness length in the WRF surface layer schemes. Like the current feedback, wave feedback to the roughness length requires change in each surface scheme, as each scheme computes roughness length locally. Here, the implementation has been performed in the Revised MM5 Monin-Obukhov scheme (Jimenez et al. 2012, sf_sfclay_physics = 1 or 91 in "namelist.input") and can be





activated using "isftcflx = 5" in "namelist.input". The changes in both "module_sf_sfclay.F" and "module_sf_sfclayrev.F" are shown in Figure 17.

```
! AHW: change roughness length, and hence the drag coefficients Ck and Cd
        IF ( PRESENT(ISFTCFLX) ) THEN
            IF ( ISFTCFLX.EQ.5 ) THEN
! isftcflx=5 is for coupling with a wave model
! varying charnock coefficient CHA_COEF from the wave model is used instead
! of CZO in the computation of roughness length ZNT
            ZNT(I)=CHA_COEF(I)*UST(I)*UST(I)/G+0.11*1.5E-5/UST(I)
        ELSEIF
            ...
        ENDIF
    ENDIF
```

**Figure 17: Modifications of "module_sf_sfclay.F" and "module_sf_sfclayrev.F" to include roughness length coupling. Lines added for the coupling interface are in blue. Original code lines are in black.**

Implementation of atmosphere-wave coupling in other surface scheme is not done yet, but as shown here, it only requires few modifications of the WRF original routines.

Implementing other ways to account for wave feedback, as using bulk formulations based on wave parameters as significant wave height, wavelength, wave age or else (e.g., Taylor and Yelland 2001, Oost et al. 2002, Warner et al. 2010, Sauvage et al. 2023) only requires few modifications in WRF original routines, similarly to what is performed here, and the addition of new coupled fields in "module_cpl.F", similarly to what is performed for CHA_COEF. Note that the MYNN surface scheme ("module_sf_mynn.F") already includes options to use parameterizations like Taylor and Yelland (2001), which estimate wave parameters from the 10m-wind speed and do not incorporate actual wave parameters. Implementing the use of mean wave parameters provided by an actual coupled wave model would therefore be straightforward in this scheme. It would only require receiving the necessary coupled fields in the coupling interface "module_cpl.F", similarly to what is performed for CHA_COEF.

### 3.3 Exchanged variables

The complete list of variables that can be exchanged between WRF and an external model is shown in Tables 3 (received variables) and 4 (sent variables). Extending the list of coupling variables if the proposed set doesn't meet the user's needs is easy. The maximum number of coupling variables to be potentially sent or received is defined by the parameter "max_cplfld" in "frame/module_driver_constants.F". Its default value (20) can be increased to any size if needed.

Each coupling variable is identified through a name, which is hardcoded and stored in a character array: either rcvname or sndname that are private variables of the module "frame/module_cpl.F". The maximum length of rcvname or sndname names is arbitrary defined to 64 characters. For code readability, and easy identification of the exchanged variables, and of the external domain (i.e. a grid domain other than WRF d01, d02...) from which / to which they are exchanged, we decided that names used to identify coupling variables must be composed of 3 parts following this convention:





1. Start with WRF_dxx, with xx a 2-digit integer specifying WRF domain number (parent or child) which sends or
receives the data.
2. Continue with _EXT_dyy, where yy is a 2-digit integer specifying the number of the external domain with which
   the exchange must be done (see section 4.1 for further details on external domain usage).
3. End with the suffix _XXX where XXX is a made of any character used to designate the field to be exchanged.

For example, if we want to exchange the sea surface temperature (identified as 'SST') between the 2nd domain of WRF and
the 3rd external domain, we will use the following name: 'WRF_d02_EXT_d03_SST'.

| Name Suffix | Description | Unit |
|---|---|---|
| SST | sea surface temperature | K |
| UOCE | ocean surface current along the x-direction | m/s |
| VOCE | ocean surface current along the y-direction | m/s |
| EOCE | eastward ocean surface current | m/s |
| NOCE | northward ocean surface current | m/s |
| CHA_COEF | Charnock coefficient used for surface fluxes computation | |

**Table 3: List of the variables potentially received, in the current version of the coupling interface.**

| Name Suffix | Description |
|---|---|
| **Mass fluxes (kg/m2/s = mm/s, positive downward)** | |
| LIQUID_PRECIP | total liquid precipitation (convective + non-convective) |
| SOLID_PRECIP | total solid precipitation (snow + hail + graupel) |
| TOTAL_EVAP | total evaporation |
| EVAP-PRECIP | net fresh water budget: evap - total precip (liquid + solid) |
| **Heat fluxes (W.m-2, positive downward)** | |
| SURF_NET_SOLAR | net surface shortwave heat flux |
| SURF_NET_LONGWAVE | net surface longwave heat flux |
| SURF_LATENT | surface latent heat flux |
| SURF_SENSIBLE | surface sensible heat flux |
| SURF_NET_NON-SOLAR | net surface non-solar heat flux (longwave + latent + sensible) |
| **Momentum fluxes (N m-2)** | |
| TAUX | surface wind stress along the x-direction |
| TAUY | surface wind stress along the y-direction |
| TAUE | eastward surface wind stress |





| TAUN | northward surface wind stress |
|---|---|
| TAUMOD | module of surface wind stress |
| **1st level wind speed (m/s)** | |
| WINDX_01 | 1st-level relative wind speed along the x-direction |
| WINDY_01 | 1st-level relative wind speed along the y-direction |
| WINDE_01 | Eastward 1st-level relative wind speed |
| WINDN_01 | Northward 1st-level relative wind speed |
| **Pressure at the air-sea interface (Pa)** | |
| PSFC | Pressure reduced at the sea level |

**Table 4: List of the variables potentially sent, in the current version of the coupling interface.**

The variables potentially sent by WRF to the coupler are either directly available in WRF (already defined in WRF registry files) or are computed based on existing variables (defined in WRF registry) before being sent to the coupler. We typically compute net solar and non-solar heat fluxes, as well as the net freshwater flux. We additionally coded several options for the vector fields to send or receive as that may need to be rotated if the local orientation of the i and j directions of the WRF grid differs from that of the external domain to which they are coupled. We therefore provide the vector fields in a common

geographic orientation (East and North components). An example of such treatment is available in the current code for the first-level wind speed (WND_E_01, WND_N_01), the surface wind stress (TAUE, TAUN), and the ocean surface currents (EOCE, NOCE). This work is done in the subroutine "cpl_snd" of "frame/module_cpl.F".

As mentioned in section 2.4.2, in the current version, all received variables are 2D surface fields used in

"phys/module_surface_driver.F", and are treated in the subroutine "cpl_rcv_sfcdrv". Adding other variables to be received is quite trivial as soon as we know how to use them in the different part of WRF code.

## 4 A multi-scale tool

After detailing the general structure of our coupling interface implementation in WRF, this section describes how we designed it in order to be (1) compatible with nested domains in WRF and/or in the models coupled to WRF, (2) as flexible

as possible in its usage, and (3) easy to maintain and to adapt to any future application.

### 4.1 Coupling mask for received variables

Our coupling interface has been especially designed to be compatible with the nesting capability available in WRF and/or in external models coupled to WRF. We have introduced the concept of a coupling mask to meet our needs in terms of coupling interface. This coupling mask will be used, for example, when coupling an atmospheric domain whose geographical extent is

greater than that of the ocean model. To do this, we will use for the SST in WRF a blend of the SST received from the ocean model over the area common to both models, and the SST read in wrflowinp_dxx over the part of the atmospheric domain





not covered by the ocean model. The coupling mask will also be used when coupling an atmospheric domain with two nested oceanic grids. In this case, the SST of the two ocean grids will have to be combined to fulfill the WRF SST field.

### 4.1.1 Coupling mask definition

The coupling mask, called CPLMASK, is between 0 and 1, where 1 corresponds to coupled points and 0 to uncoupled points. Values between 0 and 1 can be used to merge coupled and uncoupled values as described in 4.1.2 and 4.1.3. This coupling mask is defined as a 3D-array. Its third dimension, called "num_ext_model_couple_dom", is the maximum number of external domains involved in the coupling. Note that in the wrfinput file, CPLMASK appears as a 4D-array with the third dimension called "num_ext_model_couple_dom_stag" and the fourth dimension called "Time" (that is always equal to 1).

The third dimension must always exist even if we consider only one external domain in the coupling. In this case num_ext_model_couple_dom would be equal to 1. CPLMASK is declared in the file "Registry/Registry.EM_COMMON":

> state real cplmask i{ncpldom}j misc 1 z i0r "CPLMASK" "COUPLING MASK (0:VALUE FROM SST UPDATE;
> 1:VALUE FROM COUPLED OCEAN), vertical dim is number of external domains" ""

The dimension "num_ext_model_couple_dom" is declared in the file "Registry/registry.dimspec":

> dimspec ncpldom 2 namelist=num_ext_model_couple_dom z   num_ext_model_couple_dom

And the value of num_ext_model_couple_dom is defined in the "domains" section of WRF namelist through the variable "num_ext_model_couple_dom" (equal to 1 by default). This namelist variable is also defined in the file "Registry/Registry.EM_COMMON":

> rconfig integer num_ext_model_couple_dom namelist,domains 1 1 - "number of external models domains for
> coupling, used for the coupling mask" "" ""

As specified by "i0r" in the definition of CPLMASK in "Registry/Registry.EM_COMMON", this variable is added in the main input and restart files of WRF domain number xx (wrfinput_dxx and wrfrst_dxx), and is by default set to 0 everywhere. CPLMASK must therefore be modified according to the coupled configuration requested by the user.

Here we give a simple example of how to modify CPLMASK using the land category for the ocean flag in the LU_INDEX
variable of wrfinput_dxx. This example uses ncap2 which is one of the nco operators that are common tools to manipulate NetCDF files (Zender 2008).

> # modify CPLMASK based on the LU_INDEX used for ocean (here 17)
> ncap2 -O -s "CPLMASK(0,0,:,:)=LU_INDEX == 17" wrfinput_d01 wrfinput_d01

As each WRF domain (parent and children) has its own wrfinput_dxx input files, CPLMASK may differ for each WRF
domain. In the current implementation, as CPLMASK is defined in wrfinput_dxx it is fixed in time. If needed, we could imagine defining it differently, either through a time varying auxiliary input file (e.g. wrflowinp_dxx) or even by the means of a physical criterion on a given variable, e.g. the sea ice cover.





### 4.1.2 Coupling mask use

To favor a simple and generic management of exchanged variables and coupling mask, all variables received by a WRF
domain are defined over the entire domain independently of the geometry of the external domain which send them. In other
words, the coupler must interpolate the data from the external domain to the WRF domain without leaving any undefined
point. CPLMASK(:,:,nn), where nn is the index of the external domain of interest, is then used as a multiplying factor
applied to each variable received from the external domain nn.

It must be equal to 1 if only the field received from domain nn is considered, and 0 if the field from domain nn is not
considered. Fractional values, between 0 and 1, of CPLMASK can be used as weight factors to merge data received from
several external domains (different external model grids and/or input data prescribed in the wrflowinp_dxx file). The mask
value used to consider the data prescribed in wrflowinp_dxx is equal to 1 minus the sum of the CPLMASKs of all the
external domains involved in the coupling. The data received from the coupler and read in wrflowinp_dxx is then merged by
adding together all the weighted data:

505        field = sum(CPLMASK(:,:,nn)*field_rcv_nn) + (1-sum(CPLMASK(:,:,nn))*field_from_wrflowinp

For example, in an ocean-atmosphere configuration where SST for WRF domain 1 is received from 2 ocean model domains
and from the wrflowinp_d01 file, the merged SST would be:

       SST(:,:) = CPLMASK(:,:,1) * SST_received_from_external_domain_01

           + CPLMASK(:,:,2) * SST_received_from_external_domain_02

510            + (1 - (CPLMASK(:,:,1) + CPLMASK(:,:,2)) * SST_from_wrflowinput

As the coupling frequency of a given variable can be different for each external domain and can also be different from the
forcing interval of the wrflowinp_dxx file, it is necessary to store in memory each received or read field. This allows to
update the consolidated value by re-merging the fields as soon as one of them has been newly received from the coupler or
read in wrflowinp_dxx. All variables received from the coupler are thus stored in a structure, named "srcv", which is internal
to "frame/module_cpl_oasis3.F". They are, in this way, available for a merge at any time step even if the timing does not
correspond to the coupling date with a given external domain. Variables read in wrflowinp_dxx are stored in memory by
calling the subroutine "cpl_store_input" from "frame/module_cpl.F". Today, this routine deals only with the SST, which is
duplicated in the new variable SST_INPUT (declared in "Registry/Registry.EM") as soon as it is read in wrflowinp_dxx. It
would be trivial to do the same for the other variables received by WRF (such as UOCE, VOCE, and CHA_COEF). Today,
for the sake of simplicity, these variables have a default constant value that will be used when the sum of external coupled
domain CPLMASKs is not equal to 1 (see 4.1.3). This default value is defined in the "cpl_rcv_sfcdrv" subroutine of
"frame/module_cpl.F". We use 0.0185 for CHA_COEF. If not specified, this default value is equal to 0.0 (case of UOCE
and VOCE).

### 4.1.3 An example: Ocean-atmosphere coupling with nests on both sides

This example illustrates the coupling between WRF and an ocean model. Both WRF and the ocean model include a 2-way
nested domain. There are therefore 2 domains in WRF (d01 and d02) that will be coupled to 2 external domains (d01 and





d02) coming from the same executable, here named OCE. In order to represent the different cases possibly found in such coupling, we define the models' domains with different extents as follows (see Fig. 18a):

- WRF d01: 110°W-50°W, 45°S-20°N, resolution of ¼°, largest orange rectangle
- WRF d02: 95°W-55°W, 40°S-3°N, resolution of 1/12°, smallest orange rectangle
- OCE d01: 100°W-68.5°W, 43°S-10°N, resolution of 1/12°, largest cyan rectangle
- OCE d02: 90°W-69.5°W, 37°S-5°S, resolution of 1/36°, smallest cyan rectangle

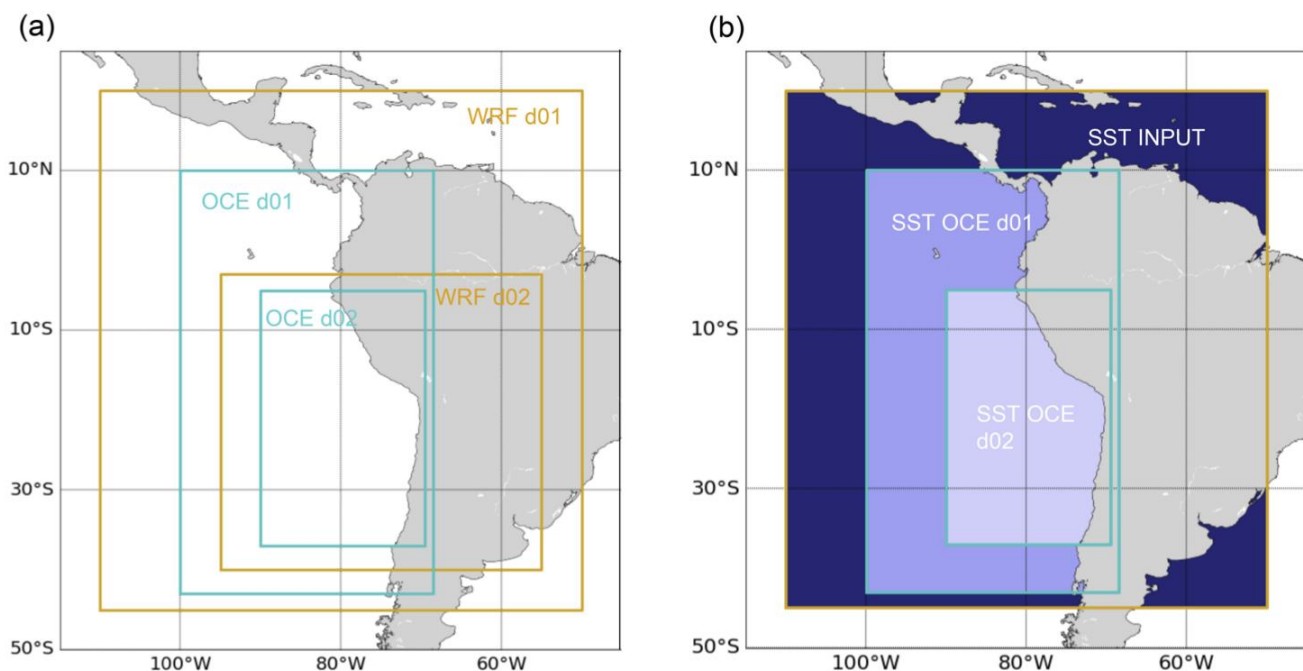

**Figure 18: Example of configuration with 2 nested domains on the atmospheric and ocean models with different extents: (a) extents of the WRF domains (WRF d01 and WRF d02, orange boxes), and of the ocean domains (OCE d01 and OCE d02, cyan boxes), (b) available sea surface temperatures (SSTs) from the wrflowinp file (identified as SST_INPUT, dark blue) and the 2 ocean domains SST (identified as SST OCE d01 in blue and SST OCE d02 in light blue).**

Let us consider the SST as an example of received field. As WRF parent domain d01 encompasses a larger region than both
the ocean domains, WRF d01 will use SST_INPUT from the wrflowinp_d01 file over places not covered by the ocean parent domain and over WRF "water points" excluded from the ocean model land-sea mask: e.g lake Maracaibo in Venezuela or the small part of the Atlantic Ocean offshore of Costa Rica, Panama and Colombia (dark blue part in Fig 18b). As OCE d01 (1/12°) is updated by the OCE d02 (1/36°) and has a finer resolution than WRF d01 (1/4°), we consider it is sufficient to send the OCE SSTs to WRF d01 only from OCE d01. The coupled mask used by WRF d01 is thus set as follows:

- The first level, CPLMASK(:,:,1), is set to 1 over the ocean part covered by OCE d01 with a 10-point wide linear transition to 0 close to the OCE d01 limits and is set to 0 elsewhere (Fig. 19a). This ensures a smooth transition between the prescribed SST_INPUT out of the OCE d01 limits and the received SST inside the OCEd01 limits.





- The second level CPLMASK(:,:,2), is 0 everywhere as we decided to not send the SST of OCE d02 to WRF d01 (Fig. 19b).

WRF child domain, d02, is almost fully covered by the ocean domains except for its southeastern corner over the Atlantic Ocean and non-ocean water bodies such as lakes and rivers/deltas, where it will thus use SST_INPUT. Over the Pacific area, both OCE d01 SST (in blue) and OCE d02 SST (in light blue) will be used (Fig 19b). The coupled mask used by WRF d02 is thus set as follows:

- The first level, CPLMASK(:,:,1), is set to 1 over the ocean part covered by OCE d01 (Fig. 19c).

- The second level, CPLMASK(:,:,2), is set to 1 over the ocean part covered by OCE d02 (Fig. 19d).

The OASIS namcouple file, which is used to set the exchanges between WRF and the OCE model in such a complex example is provided in Appendix 2.



**Figure 19: Coupling masks for the example configuration presented in Figure 18 : (a) CPLMASK for WRF d01 from external domain 1 (OCE d01), (b) CPLMASK for WRF d01 from external domain 2 (set to 0 as we do not consider the input from OCE d02 for this domain), (c) CPLMASK for WRF d02 from external domain 1 (OCE d01), (d) CPLMASK for WRF d02 from external domain 2 (OCE d02). A value of 1.0 corresponds to "fully coupled", 0.0 corresponds to "not coupled" and, for example, 0.5 to "half coupled".**





## 4.2 Deeper in the external domain concept and use

Other types of coupling could be performed with this coupling interface. We could for example imagine coupling WRF to an ensemble of N ocean models running in parallel to smooth out some part of the ocean stochastic variability. In this case, WRF executable would be coupled to N ocean executables, each one sending its own SST. In such case, we would have N

"external domain" and the weights used in each of the N coupling masks would simply be 1/N: CPLMASK(:,:,1:N) = 1./N. All the ocean models would receive the very same atmospheric forcing.

Other "unconventional" coupled experiment can be performed by extending the concept of "external domains", which are not necessarily connected to a model or an executable. In a more general view, the definition of the external domain used for sent variables even differs from the definition used for received variables.

When receiving data, the external domain can be assimilated to a single number (nn) used to identify a set of variables sharing the same coupling mask CPLMASK(:,:,nn). This means that all variables from the same external domain will be multiplied by the same CPLMASK(:,:,nn) once received by WRF. This definition usually applies to variables sent by the same domain (parent or child) of a model coupled to WRF, but other applications could be imagined.

One could decide to use different CPLMASKs for variables received from the same model. For example, if one want to test

if a specific area is key for the SST coupling, one could build a configuration where WRF is coupled to a unique ocean model sending SST, UOCE and VOCE and decide that (1) UOCE and VOCE are coupled everywhere but (2) the SST is merged with SST_INPUT (read in wrflowinp_dxx) over some parts of WRF domain. We would, in this case, need 2 coupling masks and define 2 external domains even if we couple only with 1 external model: CPLMASK(:,:,1) for UOCE and VOCE and CPLMASK(:,:,2) with a user defined geometry to merge SST and SST_INPUT where it is needed.

Conversely, one can also imagine using the same CPLMASK for variables received from different executables; implying a unique external domain definition. For example, we could imagine coupling WRF with 2 ocean models one sending the SST the other sending UOCE and VOCE and decide to use the same CPLMASK for these 3 variables even if they are sent by different models.

When sending data, the external domain can also be assimilated to a single number but, in this case, it is used to ensure the uniqueness of the name used to identify each variable sent by WRF (see naming conventions in the next section). This functionality is typically used when sending the same variable to different external domains as for example each domain could require its specific interpolation. For example, when sending the total net surface solar radiation (GSW) to the parent and child domains of an ocean model or to an ocean model and a land surface model, a different number must be associated

to each sending request of the same variable, and each receiving domain (belonging to the same model or not) must be identified with a different external domain number.

Even if we detailed here the different possibilities offered by the concept of external domain and coupling mask, we suggest, whenever it is possible, to associate each external domain number to the one of the external model domains for more clarity

when setting up a coupled configuration. Following this idea, we defined only a unique namelist parameter





"num_ext_model_couple_dom" to fix the maximum number of external domains to be considered in a simulation independently of sending or receiving action. Note that, this parameter still offers the possibility to use different numbers of external domains for sending or receiving data. Note also that "num_ext_model_couple_dom" must be smaller than or equal to the parameter "max_extdomains" defined in "frame/module_driver_constants.F". If needed, its default value (5) can be

increased to any size.

**Conclusion and Discussion**

The present paper presents the implementation and use of a non-intrusive, multi-scale, and flexible coupling interface in WRF. This interface is designed with the following baselines: (1) it is structured with a 2-level design through 2 new modules: a general coupling module, and a coupler-specific module, (2) the exchange of variables, coupling frequency, and

possible time and grid transformation are controlled thanks to an external text file, (3) a coupling mask is used to determine and potentially merge fields that can be received from various external sources (e.g., different models, domains with different resolution or wrflowinput file).

Presently, the only implemented coupler using this interface is OASIS3-MCT, but the structure is designed to allow adding more choices as the main coupling steps are thought to be generic to any coupler. The 2-level interface provides a structure

for users who wish to add their own coupler to the interface by developing the equivalent of "frame/module_cpl_oasis3.F" for their coupler. For OASIS3-MCT, all the required functionalities are likely already coded in "frame/module_cpl_oasis3.F", which should not require further modifications. This allows users to modify the coupling interface without going into the intricacies of OASIS3-MCT.

We implemented in this coupling interface the exchanges of 2D surface fields for coupling with ocean and wave models. The current list of variables potentially sent or received was detailed in section 3.3, and can be summarized as: the momentum, heat, and freshwater fluxes, as well as the adjusted sea level pressure for the possibly sent variables, and the SST, currents, and Charnock coefficient as possibly received variables. Our developed interface accounts for the vector transformation eventually needed in case of rotated grids. We also explained how coupling with surface currents or the Charnock coefficient

needs some modifications to the boundary layer and surface schemes to fully integrate the associated feedback. In WRF several options exist for such schemes, and only part of them have been modified to account for the coupling, namely the YSU and MYNN boundary layer schemes, and the Revised MM5 Monin-Obukhov surface scheme.

The coupling interface described here presently involves only 2D-arrays from the surface module, but it is completely open

and possible to implement others kind of coupling like for example with another (global) atmospheric model, or with a chemistry model that would require to couple 3D fields from other parts of the model. Adapting the current coupling interface would just requires to modify the sending/receiving routines of "frame/module_cpl.F" to be compatible with 3-D arrays, and to add calls in the WRF routines requesting the exchanged fields as we did in the surface driver with the



subroutine "cpl_rcv_sfcdrv" (see for example Briant et al. 2017). OASIS limitations would however impose to use the same

mask and the same the number/frequency of exchanged variables for every vertical level.

OASIS3-MCT requires that all the coupling specifications (i.e., which variables are sent/received to/from which domain) are defined before any coupling exchange of variables from parent grid. This constraint has two consequences. First, as explained in section 2.4.1, the coupling interface works with a maximum of one level of nested grids. However, this

limitation is relatively easy to circumvent. To do so, one would have to move the loop defining the child grids (DO WHILE ( nests_to_open… ) from integrate to "wrf_init". As this loop must be called recursively to initiate all domains, the simplest modification would be to put this loop in a small recursive subroutine called at the end of "wrf_init". Thereby, all nested grid definition would be done at once during the initialization phase, before the first time step of the parent grid in agreement with OASIS3-MCT requirements. This modification of WRF code would however prevents to instantiate and cease the child

grids at any time, a feature required by some users. Its clean implementation would therefore need additional modifications of the code to reconcile all WRF usages. We though that there is not enough need for multi-level nesting in couple mode to implement this work and decided to keep the code as closest as possible to its original version.

Another similar partial solution is to allow to have more than one nest, but to couple only the parent grid and the first nest. This can easily be achieved by limiting the number of WRF domains involved in the coupling to 2, a

modification concerning only 3 lines of code:

 - in frame/module_cpl_oasis3.F, replace

   IF ( pgrid%id == pgrid%max_dom ) CALL cpl_oasis_enddef()

by

   IF ( pgrid%id == MIN(2,pgrid%max_dom) ) CALL cpl_oasis_enddef()

- in frame/module_integrate.F, replace

   IF (coupler_on) CALL cpl_defdomain( new_nest )

by

   IF ( coupler_on .AND. new_nest%id <= MIN(2, new_nest%max_dom) ) CALL cpl_defdomain( new_nest )

and replace

IF (coupler_on) CALL cpl_defdomain( new_nest )

by

   IF ( coupler_on .AND. new_nest%id <= MIN(2, new_nest%max_dom) ) CALL cpl_defdomain( new_nest )

Second, the grid is defined only once at the initialization stage, which prevent to use the moving nest ability of WRF as the

model grid is moving over time when this option is activated. A partial solution is simply to couple only the WRF parent static domain to an ocean or wave model. The moving grids are not directly involved in the coupling interface but do play an indirect role through the feedbacks: (1) of the fields computed in WRF moving nest domain(s) to WRF parent domain, (2) of the coupled fields received in WRF parent domain and provided to the moving nest(s) (e.g.. SST). This strategy has the



advantage of using atmospheric fields calculated at high resolution in the nests and interpolated on the WRF d01 domain to feed the ocean or wave model, and of considering the feedback of surface conditions provided by the coupler in the evolution of the parent and moving nests, while maintaining the coupling interface simple. To do so, one must use 2-way moving nest(s) and allow the interpolation of WRF d01 received SST field into the moving nest, so that coupled SST is accounted for in the moving nests. This requires to slightly modify the "Registry/Registry.EM" as:

```
state   real   SST          ij   misc   1   -   i01245rh05d=(interp_mask_field:lu_index,iswater)f=(p2c)
675        "SST"        "SEA SURFACE TEMPERATURE" "K"
```

Finally, note that, WRF adaptive time-step was not tested. It probably doesn't work as, for example, the coupling time must correspond to a multiple of the time step. We also haven't tested this coupling interface in OpenMP. Although OASIS exchange routines can be used on an OpenMP-compatible model, assuming that the coupling library has also been compiled
with the OpenMP option, the coupling variables should necessarily be collected on the main thread before being supplied as arguments.

**Appendices**

**Appendix 1: shell script used to build OASIS3-MCT files, "grids.nc", "masks.nc" and "areas.nc"**

We provide here: https://github.com/massonseb/WRF/blob/GMD_wrf_coupling/tools/create_wrf_grids_masks_areas.sh, a
shell script that can be used to build the OASIS3-MCT files called "grids.nc", "masks.nc" and "areas.nc" from the WPS geogrid file. Note that this shell script uses nco operators (Zender 2008).

**Appendix 2: example of a namcouple file**

We provide here: https://github.com/massonseb/WRF/blob/GMD_wrf_coupling/run/namcouple_example, as an example, the namcouple file used in the coupled model described in section 4.1.3.

**Code availability**

This paper refers to a modified version of WRF v4.6.0 that is available on GitHub at the following address https://github.com/massonseb/WRF/tree/GMD_wrf_coupling or through Zenodo DOI: https://doi.org/10.5281/zenodo.13350615

**Author contribution**

S. Masson developed this OASIS3-MCT-WRF coupling interface with the help of E. Maisonnave (for the OASIS3-MCT interface), D. Gill (for WRF interface and particularly the nesting capacity) and G. Samson. S. Jullien, M. Le Corre worked





on the wave coupling part. S. Masson tested this coupling interface with the ocean model NEMO. S. Jullien, M. Le Corre and L. Renault tested it with the ocean model CROCO. This paper was mainly written by Sébastien Masson and Swen Jullien with the inputs and the comments of the other co-authors.

**Competing interests**

The contact author has declared that none of the authors has any competing interests.

**Acknowledgements**

S. Masson would like to thank the "Capacity Center for Climate & Weather Extremes" team from the "Mesoscale & Microscale Meteorology" laboratory at NCAR for welcoming him in July 2013 and 2014 to work on the original version of this coupling interface. This work was supported by the ANR project Peta scale mULti-gridS ocean-ATmosphere coupled simulatIONS (PULSATION, ANR-11-MONU-0010).

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
