# Peer review of "An updated non-intrusive, multi-scale, and flexible coupling interface in WRF 4.6.0"

_Geoscientific Model Development, 2024_

## Author Comment (AC1)

**Review 2:**

1. The mention of the "most popular model" regarding WRF appears subjective (Short summary, and beginning); unless this is proven, suggest rephrasing L38: "among the most popular…" in both occurrences.

We have corrected the text as proposed by the reviewer.

2. L38: not convinced that "numerical performances" are among the main reasons for the success of WRF. Unless the latest versions have changed this, WRF is known for limited scalability.

We agree, we have suppressed this reference to WRF numerical performances.

3. L61 please introduce the acronym C-Coupler used later

We corrected that, when introducing the Community Coupler: *"Community Coupler (C-Coupler, Liu et al, 2014)"*. We also added the missing reference for YAC and repeated C-Coupler reference, line 359.

4. L83 the sentence "The de facto" needs to be rephrased for clarity

We agree, we have removed this sentence that was repeating information already mentioned in the previous/following sentences.

5. Section 2.3 refers to the standard OASIS functionalities. This needs to be stated clearly; also, the associated schemes (fig3) and text do not consider any delay (LAG) in the time-stepping to exchange fields and avoid deadlock communication. I recommend adding a discussion on that, which could be useful for many users.

We have added "Following OASIS3-MCT approach" at the beginning of the first sentence in Section 2.3.
We have also added the following paragraph at the end of section 2.2 (lines 167-185):

*In forced mode, WRF reads the surface boundary conditions at the beginning of the time step. In coupled mode, these quantities are provided by the coupler. In the « real world » the air-sea exchanges are continuous, but it is not easy to achieve such synchronicity in coupled models. The cleanest way would be to iterate the coupling procedure several times at each coupling time window until flux computation converges (Lemarié 2008). The computational cost of this methodology is, however, prohibitive. A compromise could be to have a coupling time step small enough to represent the continuous air-sea exchanges. This solution is often*

*not compatible with the relatively large time step of ocean models and the uncertain validity over small time windows (< 10 minutes) of bulk formulations that have been calibrated using average hourly measurements (Large, 2006). The usual solution is to exchange averaged fields over a coupling time window that is considered "small enough" to represent a kind of synchronicity while being compatible with the ocean model time step and the bulk formulations. To obtain the best compromise between numerical performance and the coherence of the coupling fields, the ocean and atmospheric models run in parallel rather than sequentially (see figure 4 of Valcke et al. 2013) using dedicated MPI resources (see details on MPI communication in section 2.4 and MPI resource allocation in section 2.6). The atmosphere modifies the ocean state which will not feedback to the atmosphere immediately but with a delay of the coupling time window. In OASIS3-MCT, a functionality called "lag" is used to synchronize the send and receive functions and avoid deadlock between models (detailed in section 2.5.3 of the OASIS3-MCT user guide, Valcke et al. 2021). In our implementation, the sending function is called at the end of the time step, and the receiving function is called at the beginning. Synchronous exchanges therefore require sending to take place during the time step preceding reception, which is ensured by defining the lag to one time step of the sending model. Note that at the beginning of the simulation, the variables to be received are read in NetCDF restart files written by the sending models at the end of the previous simulation.*

6. L200 decision -> strategy
Corrected.

7. In Table 2, every interpolation method is given also with the "F" (BICUBIC and BICUBICF) without explaining the meaning
We have now specified in the text of the table: "with (BILINEAR) / without (BILINEARNF)" and repeated these modifications for BICUBIC, DISTWGT and GAUSWGT. We have also specified LOCCUNIF, LOCCDIST and LOCCGAUS definitions.

8. L281 as shown in Figure 9.
Corrected.

9. It is not clear why the use of relative wind is implemented only in two PBL schemes, and not in all; is there a specific reason? Can the authors at least say how to do it for the other schemes?
The modifications for the relative wind were done in the two (most) popular PBL schemes. We have modified the sentence line 412:

*"The implementation of the requested modifications has, for now, been done in 2 popular PBL schemes: the Yonsei University (YSU, Hong et al. 2006) and the Mellor–Yamada Nakanishi Niino (MYNN, Nakanishi, M., and H. Niino, 2009) schemes."*

We have added a reference to a paper from Samelson et al., in which they modified other PBL schemes (line 414):

*"Samelson et al. 2014 explored the impact of the relative wind in other PBL schemes and proposed the corresponding WRF modifications in the GitHub repository associated with the publication."*

10. The same applies to the Charnock coefficient in the Revised MM5 Monin-Obukhov surface scheme, which could be discussed in more detail. What about other schemes?

The use of a Charnock coefficient from a wave model was added as one of the options of the namelist parameter "isftcflx" which is described in the README.namelist as "alternative Ck, Cd formulation for tropical storm application". In WRF 4.6.0, the "isftcflx" parameter is implemented only in the surface schemes 1, 91 and 5 which corresponds to the revised and the old MM5 and the MYNN surface layer schemes. Following the suggestion of reviewer 2 and what was already done for "isftcflx", we added the possibility to use a wave model Charnock coefficient in MYNN surface layer scheme.

We modified "module_sf_mynn.F", the caption of figure 17 and the following paragraph (lines 434-442):

*Here, the implementation has been performed in the 3 schemes that use the "isftcflx" namelist parameter defining alternative Cd formulation: the Revised MM5 Monin-Obukhov scheme (Jimenez et al. 2012, sf_sfclay_physics = 1 or 91 in "namelist.input") and the MYMM scheme (Olson et al. 2021, sf_sfclay_physics = 5 in "namelist.input"). Coupling through the exchange of the Charnock coefficient is activated by using "isftcflx = 5". The changes made in "module_sf_mynn.F", "module_sf_sfclay.F" and "module_sf_sfclayrev.F" are shown in Figure 17.*

11. L631 "other kinds"

Corrected.

12. In general, the technical paper will be very useful if associated with realistic examples collected by the authors and the community (in terms of masks, grids, namelists). The example provided in the github is quite limited (exchanges of only SST and TAUX); having at least two complete configuration examples (e.g., one with no nested domains and one with), complete as in realistic applications of all domain and grid files, the scripts to

generate them and the associated namelists to run, could be very useful to guide users in their implementations.

We agree with the reviewer and create two Zenodo repositories containing all input files to run two different examples. We added the following lines at the end the Appendix 2.

*Input files for running real applications are also provided in two Zenodo repositories. The first one is an example of a coupling between WRF, WAVEWATCH III and CROCO (https://zenodo.org/records/14235410), the second one is an example of a WRF-CROCO coupling with a 2-way nested domain in CROCO (https://zenodo.org/records/14235450).*

---

## Author Comment (AC2)

**Review 1:**

1. For the implementations described in Section 2, the implemented interface is specifically developed for WRF v4.6.0. If so, are there any plans to maintain this interface for higher or lower versions of WRF? Would it be easy to implement this interface for other WRF versions?

As explained in the paper (lines 69-71), this coupling interface is an update of the coupling interface already available in WRF since 2014. All the changes made to the code are available on GitHub. They are listed here https://github.com/massonseb/WRF/commits/GMD_wrf_coupling/?author=massonseb

As the interface is non-intrusive, it should not necessitate work to maintain it in past/future WRF versions. Apart from the changes we have made for wave coupling, our update of the coupling interface concerns module_cpl.F, the replacement of 3 calls to cpl_rcv by a single call to cpl_rcv_sfcdrv in module_surface_driver.F and the use of the variables cosa and sina in module_first_rk_step_part1.F and module_surface_driver.F. Backporting these changes to older versions of WRF will therefore be fairly straightforward. We are currently preparing a "merge request" on GitHub to integrate this update to the coupling interface into the official WRF repository, which would ensure its use in future versions.

We added the following sentence at the end of the paragraph, line 72:

*Note that all changes made to the code are available on github at the following address https://github.com/massonseb/WRF/commits/GMD_wrf_coupling/?author=massonseb. They are limited to a few routines, so porting them to older versions of WRF will be fairly straightforward.*

2. Line 155 is confusing to me. Do the authors separate the WRF code into three subroutines (init, run, finalize)? Or do the authors put functions in "frame/module_cpl_oasis3.F" used in these processes?

We did not separate the WRF code into 3 subroutines. The coupling interface provides (in "frame/module_cpl_oasis3.F") subroutines that will be used in the native WRF processes how it is illustrated in Fig. 4. We added *"provided in the coupling interface"* in the following sentence line 155:

*The coupling sequence (Fig. 2) is structured into 3 steps with functionalities corresponding to specific Fortran subroutines provided in the coupling interface and callable from the original code*

3. Are the authors using the existing WRF I/O streams (auxinput or auxhist) when getting the input or output? Why don't use the existing ones?

This coupling interface does not interfere with existing WRF I/O streams (auxinput or auxhist). Auxinput and auxhist files are used the same way with or without activating the coupling interface at the compilation stage (see section 2.5). This interface is also compatible with WRF IO quilting (see lines 258-259) and we used WRF IO quilting in several of our coupled simulations. We also clarified the use of WRF IO quilting in the new section 2.6 (lines 382-393).

Having said that and regardless of the coupling interface, in our forced or coupled simulations, we use WRF auxinput stream, but we don't generally use the auxhist stream. Instead, we use the XIOS I/O server (https://forge.ipsl.jussieu.fr/ioserver/wiki) to produce the output files. Our main reason for using XIOS is that it can calculate the time average/max/average/std of any variable declared in the Registry without changing anything in the code. We find it much more convenient than modifying the WRF subroutine clwrf_output_calc in phys/module_diag_cl.F. We find XIOS is much more flexible than the default WRF output stream. We also think XIOS is more efficient than WRF IO quilting. The use of the XIOS-WRF is not related to the coupling interface described in this paper, and we don't think we should mention it.

4. Line 236, when in the coupled experiment. I feel it is challenging to use WRF IO quilting when using the same processors for both Ocean and Atmosphere models. How should the ocean model set these processors for IO quilting? Have the authors tested this?

In this question, does the term "processor" refer to a CPU made up of several cores or to an MPI process?
In this coupling interface, the ocean and atmosphere models run on different and dedicated MPI processes and not sequentially on the same MPI process.
We added a new section "2.6 Running the coupled model" to detail and clarify how to run and allocate MPI resources to the coupled model (lines 382-393):
*"To achieve higher parallelism and keep the different models as independent as possible, each model has its own executable with its dedicated MPI resources. All the executables run in parallel and share the same MPI world using the "multiple programs, multiple data" (MPMD) launch mode. WRF can be coupled to one or several external models. WRF and the external models can include one or several domains (e.g. embedded zooms, see section 4). For example: if X is the number MPI tasks allocated to WRF and Y the number of MPI tasks allocated to the external model to which WRF is coupled (e.g. an ocean model), the number of MPI tasks that must be allocated to run the simulation is X + Y. Then, if using WRF IO quilting, the X WRF MPI tasks will be split among X1 "compute nodes" and X2 "server nodes" with X = X1 + X2. If WRF is configured with one nested zoom, the parent and the child domains, d01 and d02, will run sequentially on X1 MPI tasks. The same applies to the external model, which could include its own zoom domains running on Y MPI tasks. Note that the external models do not use/see WRF I/O quilting. OASIS3-MCT can couple models sharing the same executable, so we could imagine to further integrate WRF I/O quilting in the coupling interface, but this would require specific modifications of the external models and greater entanglement between the different codes, which is not our objective."*

5. Section 3.3. Are the users free to add more variables for the coupling processes?

Yes definitely, this is one of the aims of this interface. We designed this interface (isolating the coupling interface in module_cpl.F and the OASIS-specific routines in module_cpl_oasis3.F) to make changes as easy as possible. Such procedure is detailed in section 3.3.

6. Section 4.1.3 is a very interesting example. Has anyone tested the nested domains in a realistic application? In addition, in Fig. 19 the filled color and the color of the boxes are very close. It would be better if the author could use a different colormap for the masks.

Yes, we have used this coupling configuration in realistic configurations (tropical channel, South-East Pacific, Gulf Stream...) with nested atmospheric and/or oceanic zooms for more than 10 years. See for example this presentation of unpublished work, https://www.clivar.org/sites/default/files/documents/wgomd/ws2014/04__highres_Masson.pdf

It is true that most existing publications using this coupling interface do not use atmospheric zooms, with the exception of:

Li, Y., Jourdain, N. C., Taschetto, A. S., Gupta, A. S., Argüeso, D., Masson, S., and Cai, W.: Resolution dependence of the simulated precipitation and diurnal cycle over the Maritime Continent, Clim Dyn, 48, 4009–4028, https://doi.org/10.1007/s00382-016-3317-y, 2017.

Figure 19 have been changed according to the reviewer's suggestions:

[Figure]

7. In Appendix 2, there is a typo in https://github.com/massonseb/WRF/blob/GMD_wrf_coupling/run/namecouple_example. It should be https://github.com/massonseb/WRF/blob/GMD_wrf_coupling/run/namcouple_example.

Thank you, we corrected it.